# A Fast Learning Method for Accurate and Robust Lane Detection Using Two-Stage Feature Extraction with YOLO v3

**DOI:** 10.3390/s18124308

**Published:** 2018-12-06

**Authors:** Xiang Zhang, Wei Yang, Xiaolin Tang, Jie Liu

**Affiliations:** 1State Key Laboratory of Mechanical Transmission, College of Mechanical Engineering, Chongqing University, Chongqing 400044, China; zkebi@126.com; 2State Key Laboratory of Mechanical Transmission, College of Automotive Engineering, Chongqing University, Chongqing 400044, China; Knight_jie2018@163.com

**Keywords:** lane detection, YOLO v3, adaptive learning, label image generation

## Abstract

To improve the accuracy of lane detection in complex scenarios, an adaptive lane feature learning algorithm which can automatically learn the features of a lane in various scenarios is proposed. First, a two-stage learning network based on the YOLO v3 (You Only Look Once, v3) is constructed. The structural parameters of the YOLO v3 algorithm are modified to make it more suitable for lane detection. To improve the training efficiency, a method for automatic generation of the lane label images in a simple scenario, which provides label data for the training of the first-stage network, is proposed. Then, an adaptive edge detection algorithm based on the Canny operator is used to relocate the lane detected by the first-stage model. Furthermore, the unrecognized lanes are shielded to avoid interference in subsequent model training. Then, the images processed by the above method are used as label data for the training of the second-stage model. The experiment was carried out on the KITTI and Caltech datasets, and the results showed that the accuracy and speed of the second-stage model reached a high level.

## 1. Introduction

To meet the needs of intelligent driving, the lane detection algorithm must have high accuracy and real-time response [1,2]. Due to the variety of the driving environment, the fast and accurate detection of a lane under complex road conditions is always a great challenge. Therefore, lane detection has attracted the high attention of researchers in the lane detection field. Low et al. [3] fitted the lane by detecting the edge of a lane using Canny operators, finding the best line using the Hough transform. Bounini et al. [4] proposed an algorithm for road boundary and lane detection based on the Hough transform, which combines the Canny edge detector, least square method, and Kalman filter to predict the location of a road boundary. Ding et al. [5] proposed a vision-based road ROI (Region of Interest) determination algorithm, which uses the information on the location of vanishing points to detect road regions, and finally uses the Hough transform to detect line segments. El Hajjouji et al. [6] divided the lane detection into two parts. First, the adaptive threshold Sobel operator and Hough transform were used to detect a lane which was then tracked by Kalman filter. However, the method for lane detection based on the Hough transform has the shortcoming that, when there are many lines in the image, the false rate of a lane is relatively high [7,8].

To improve the accuracy of lane detection, de Paula and Jung [9] proposed a lane detection algorithm based on a generated top-view image, by constructing a curve road model, the detected lane can be extended to the bottom of the region of interest (ROI). Chi et al. [10] used a linear parabola lane model to detect a lane boundary, and a Bayesian classifier based on the Gaussian Mixture was employed to classify the lane which was marked as dotted, solid, or dotted-solid line respectively. Revilloud et al. [11] proposed a lane detection algorithm based on the vanishing point estimation. First, the location of a lane vanishing point was determined in the current frame, and then the region under the vertical coordinate of the vanishing point was taken as an ROI. After completing the detection of the edge point in the ROI, the pixels at the edge points were fitted by the Hough transform [12], and then, the longest line in the range of preset angles was taken as a lane. These methods can achieve better results on the roads with smaller color change, but the detection accuracy is not high when the brightness changes under the condition of backlighting, shadow, and rainy weather.

To reduce the impact of the environmental factors on the detection algorithm, some researchers used a LiDAR (Light Detection And Ranging) to detect lanes. Shin et al. [13] proposed an algorithm for lane detection using both camera and LiDAR, and the ability of lane detection in a harsh environment was improved by employing the images and LiDAR information. Rose et al. [14] estimated the distance between the vehicle and lateral lane using the information from cameras, GPS, and LiDAR; the recognition accuracy reached the centimeter level. Although the LiDAR has low sensitivity to the environment, its price is high, so it is difficult to achieve a large-scale application in the lane detection systems.

With the rapid development of the deep learning and other related technologies, the CNNs (Convolutional Neural Networks) [15] have shown significant advantages in the extraction of image features, so they have been widely used in object detection and classification. In studying lane detection, some scholars have designed CNNs with different depth, and trained the deep CNN models using label images. Brust et al. [16] used a CNN to learn the features of lanes, and combining the spatial information of image improved the accuracy of lane detection. Pan et al. [17] presented a new network named the Space CNN, where using the feature mapping in the convolutional layers, the pixel information transfer in rows and columns between layers was realized. Such a network structure is suitable for detecting the objects with a strong spatial relationship, such as the lane. As it is easier for vehicles to identify lanes in the traffic scenarios, Satzoda and Trivedi [18] first trained a CNN model for vehicle detection, and then used the information on vehicle location to constrain the detection range of lanes, improving the accuracy of lane detection. Kim and Lee [19] first identified the complexity of the scenario, and in simple scenarios, only the RANSAC algorithm [20] was used to detect a lane, and in the complex scenarios, a CNN was first used to extract the lane features, and then, the RANSAC was used to detect the lane. Li et al. [21] used a CNN to extract the information on lane feature first, and then trained a recurrent neural network using the spatial relationship between lanes, which improved the detection accuracy.

The above algorithms for lane detection based on the CNNs generally train a network composed of only a few convolution layers and pooling layers, the detection accuracy and speed are generally not satisfactory. With the rapid development of object detection technology in recent years, the speed and accuracy of object detection technology have been greatly improved [22]. Therefore, in this work, a lane detection model based on the object detection algorithm that takes advantage of high accuracy and fast processing speed of the object detection algorithm to complete the lane detection and tracking is introduced. In the field of object detection, a common solution is to retrain the existing classifiers and assign labels to the object bounding box. For instance, using the standard sliding window method, the position and category of the object can be determined by scanning the whole image [23]. However, this method has a high false detection rate and high computational complexity, which is not conducive to the real-time object detection. Both Region-CNN (RCNN) [24] and Fast RCNN [25] use the selective search algorithm to select the candidate bounding boxes. The Faster RCNN [26] is based on the idea of obtaining a candidate box by an RPN (Region Proposal Network). However, object detection in these methods requires two-step processing, including positioning and object classification, which has a slow arithmetic speed, so it is difficult to realize real-time detection.

The YOLO (You Only Look Once) algorithm proposed by Redmon et al. [27] classifies and locates the object in only one step and obtains the position and category of the object directly at the output layer. In YOLO, object detection is treated as a regression problem, which greatly improves the operation speed by meeting the real-time requirements. The SSD (Single Shot multibox Detector) [28] integrates the end-to-end idea of the YOLO and the anchor boxes mechanism of the Faster RCNN, improving the detection ability for small objects by increasing the size of the input image. The YOLO v2 [29] adds anchor boxes on the basis of the YOLO and trains the bounding boxes to find better dimensions of boxes automatically using the K-means [30], which makes it easier and more accurate to predict the object location. However, since less information on small objects is saved in the high-level feature map, when there are too many adjacent objects, the detection result is not satisfactory. The YOLO v3 [31] uses the network adapting features on the basis of Darknet-53, and the softmax loss in the YOLO v2 is replaced by a logistic loss, which has obvious advantages in small object detection.

Due to the advantages of YOLO in object detection, many scholars have carried out object detection in many applications by using YOLO as a detector, with satisfying results. Romain et al. [32] produced snippet action proposals by combining the regression-based RNN detector and YOLO. Then, these short action proposals are integrated to form final action proposals. Alvar and Bajić [33] constructed a fast and efficient tracking engine based on YOLO, and the results on the OTB tracking dataset indicate advantages of the proposed method in terms of accuracy and speed. Brilian et al. [34] used YOLO as the object detector and polynomial regression as the road guidance in the real-world driving video simulations, and the self-driving car was developed. Ning et al. [35] developed an approach of spatially supervised recurrent convolutional neural networks based on YOLO for visual object tracking, it can exploit the history of locations as well as the distinctive visual features.

This paper aims to obtain the lane position on a road. A lane size is relatively small compared to the whole image, and the lane spacing becomes shorter after it is converted to the bird-view image. From the above analysis, the YOLO v3 has obvious advantages in detecting small target objects regarding the speed and accuracy. Therefore, based on the YOLO v3 framework, a lane detection model is designed, and the lane detection and tracking are completed.

The training of a deep CNN model requires a large number of label images. As there are few standard lane label datasets available, manual labeling is required. A simple method for label image generation is the time-slice, wherein the lane position is labeled in the successive images frame by frame, but the labeling process is tedious and time-consuming, and the label data is affected by the subjective factors which may lead to the inaccurate label [36]. In this paper, an algorithm for the automatic generation of label images is proposed, which provides accurately labeled data for model training, and improves the training efficiency of the model significantly.

The main contributions of this work are as follows.
A method for automatic generation of the lane label images is presented. The lane is automatically labeled on the image of a simple scenario using the color information. The generated label dataset can be used for the CNN model training.A lane detection model based on the YOLO v3 is proposed. By designing a two-stage network that can learn the lane features automatically and adaptively under the complex traffic scenarios, and finally a detection model that can detect lane fast and accurate is obtained.As the lanes detected by the network model based on the YOLO v3 is relatively independent, the RANSAC algorithm is adopted to fit the final required curve.

The paper is organized as follows. In Section 2, a method for generation of a label image is presented to provide the data for the training of the first-stage model, and then, a two-stage network is constructed and the method for adaptive learning of lane features is introduced in detail. In Section 3, the effectiveness of the proposed detection model is evaluated by experiment. In Section 4, the results are summarized, and the main conclusions are given.

## 2. Lane Detection Algorithm

### 2.1. Automatic Generation of Label Images

For CNN model training, a large number of label images is required. However, the acquisition of the label images is still challenging. Currently, there are fewer standard label images for lane detection and tracking, so we propose a method for automatic generation of label images by which the lane can be identified in a simple scenario, and the location of the lane can be determined accurately.

Before lane detection, the camera is generally installed inside the front windshield, so the captured images contain information that is not related to the lane, such as sky, vehicles, trees by the roadside, and so on. Through the warp perspective mapping (WPM) [37] of the images collected by the camera, the obtained bird-view images mainly contain the road surface and lane, further, in the bird-view images, the lane appear in the form of parallel lines, which makes it more convenient for subsequent processing.

To mark a lane in a bird-view image, the information on lane coordinates should be extracted first. The lanes on the roads are usually in white and yellow colors. Since L channel of the *LUV* color space [38] and *B* channel of the *LAB* color space [39] are sensitive to the white and yellow colors, respectively, the *LUV* and *LAB* are used in this study to extract white and yellow lines, respectively. First, by setting the minimum threshold *l_thresh_min* and the maximum threshold *l_thresh_max* for *L* channel of the *LUV*, the value of *L* channel in the image between these two thresholds is labeled as a white lane. Similarly, for yellow lanes, the minimum threshold *b_thresh_min* and the maximum threshold *b_thresh_max* are set for B channel of the LAB. After that, the white and yellow lines are binarized, and then, the connected domain is detected to determine the coordinates of each lane in the image.

### 2.2. Construction of Detection Models

The proposed two-stage network aiming to learn the lane features adaptively is shown in Figure 1. Firstly, a first-stage lane detection model based on the YOLO v3 is trained using the label images which are generated as explained in Section 2.1. The obtained first-stage model can predict the lane location in a simple scenario. To improve the detection ability of the model in the complex scenarios, the method of transfer learning [40] is adopted in the second-stage training, and the raw unlabeled images are used to re-train the primary model, thereby enabling the resulting model to identify a lane in complex scenarios.

During the detection using the YOLO v3, the image is divided into *S × S* grids, and the candidate boxes are distributed with the same density on the *x* and *y* axes. In fact, the lanes are sparsely distributed on the x-axis and relatively dense on the *y*-axis in the bird-view image, as shown in Figure 2. To mitigate the influence of the aspect ratio on object detection, the structure of the detector is improved by dividing the image into *S* × 2*S* grids to increase the longitudinal detection density, making it more suitable for lane detection. Each grid predicts whether the center of a lane falls into its interior. If the prediction is positive, then the grid predicts B bounding boxes and the confidence of each box. The confidence denotes the confidence degree of the detection bounding box to detect the object, and it is defined by:(1)Conf(Object)=Pr(Object)×IOU(Pred,Truth),
where *Pr* (*Object*) indicates whether there is a lane in the predicted bounding box (1 for yes, 0 for no), and *IOU* (*Pred, Truth*) is the intersection of the real box and the predicted box, and it is defined by:(2)$$IOU\left(Pred,Truth\right)=\frac{area\left(Box_{T}\cap Box_{P}\right)}{area\left(Box_{T}\cup Box_{P}\right)},$$
where *Box_T_* is the ground truth box based on the training label, *Box_P_* is the predicted bounding box, and *area*(·) represents the area of intersection.

In the lane detection process, the suitable candidate areas are selected and predicted first. Then, the prediction results are screened, and the predicted boxes with high confidence are obtained. Each bounding box contains five predictive values: *x*, *y*, *w*, *h*, and confidence, *ξ*, among which (*x*, *y*) denote the coordinates of the bounding box center, and *w* and *h* are the width and height, respectively. After obtaining the confidence *ξ* of each prediction box, the boxes with a low score is removed by setting a threshold, and then the remaining bounding boxes are processed by the non-maximal suppression (NMS) to obtain multiple sets of high-score bounding boxes, and finally, the position parameters are derived.

To calculate the loss, the error of the predicted coordinates and bounding box of lane, and the classification error need to be determined. Since lane is the only object in this study, there is no classification error, so the loss function of the model can be expressed by: (3)$$loss\left(lane\right)=E_{c}+E_{IOU},$$
where $E_{c}$ and $E_{IOU}$ are the errors of the coordinates and bounding box between the predicted and the ground truth data, respectively. 

The size of a lane in an image can be different, but the loss function of the YOLO v3 takes the same error for all boxes; however, the error of a large- and small-size objects has different effects on the whole image. Therefore, the loss function of the coordinate error is improved by using the contrast normalized. The loss function of coordinate error $E_{c}$ is given by:(4)$$E_{c}=\lambda_{coord}\sum_{i=0}^{2S^{2}}\sum_{j=0}^{B}I_{ij}^{lane}\left[{\left(x_{i}-{\hat{x}}_{i}\right)}^{2}+{\left(y_{i}-{\hat{y}}_{i}\right)}^{2}\right]+\lambda_{coord}\sum_{i=0}^{2S^{2}}\sum_{j=0}^{B}I_{ij}^{lane}\left[{\left(\frac{w_{i}-{\hat{w}}_{i}}{{\hat{w}}_{i}}\right)}^{2}+{\left(\frac{h_{i}-{\hat{h}}_{i}}{{\hat{h}}_{i}}\right)}^{2}\right],$$
where $\lambda_{coord}$ is the weight coefficient of coordinate error that emphasizes more on the coordinate training and here $\lambda_{coord}=5$ is referenced from [27]; $x_{i}$, $y_{i}$, $w_{i}$, and $h_{i}$ represent the center coordinates, length and width of the predicted *i*th cell, respectively; ${\hat{x}}_{i}$, ${\hat{y}}_{i}$, ${\hat{w}}_{i}$, and ${\hat{h}}_{i}$ represent similar parameters of the real *i*th cell, respectively; $I_{ij}^{lane}$ represents the possibility of a lane in the *j*th bounding box of the *i*th grid.

The $E_{IOU}$ error of $IOU_{pred}^{truth}$ can be calculated by:(5)$$E_{IOU}=\sum_{i=0}^{2S^{2}}\sum_{j=0}^{B}I_{ij}^{lane}{\left(\xi_{i}-{\hat{\xi }}_{i}\right)}^{2}+\lambda_{nolane}\sum_{i=0}^{2S^{2}}\sum_{j=0}^{B}I_{ij}^{nolane}{\left(\xi_{i}-{\hat{\xi }}_{i}\right)}^{2},$$
where $\lambda_{nolane}$ represents the weight coefficient of the classification error, $\xi_{i}$ is the confidence value of the *i*th cell in the predicted sliding window, ${\hat{\xi }}_{i}$ represents the confidence value of the *i*th cell in the real sliding window. $I_{ij}^{lane}$ denotes whether the lane appears in *j*th bounding box in cell *i*, and if it appears, the value of $I_{ij}^{lane}$ is 1, otherwise it is 0. On the contrary, if there is no lane in in *j*th bounding box in cell *i*, the value of $I_{ij}^{nolane}$ was set as 1, and if otherwise it is 0. Considering that $IOU_{pred}^{truth}$ error of a cell with and without detection object differs, the contribution to the loss function of the whole training process is also different. If the same weight is used, the confidence error of a cell containing the detected object will be larger. Therefore, $\lambda_{nolane}=0.5$ is used in the YOLO v3 to reduce the transfer error.

### 2.3. Adaptive Learning of Lane Features

Since the label images are all obtained in simple traffic scenarios, the model trained in the first-stage is suitable only for simple scenarios. To improve the generalization ability of the model and make the model able to deal with the lane detection under complex conditions, a framework for adaptive and learning of lane features, which automatically learns the lane features in different scenarios, is proposed. 

During the training of the second-stage model, the larger the *ξ*, the greater the confidence that the lane is detected. By setting a threshold *T*, if *T ≤ ξ*, it is considered that the lane is detected. The YOLO v3 model in the first stage can output the lane coordinates, but these coordinates cannot be used directly for model training because the position of lanes are not accurate enough, so it is necessary to re-position the lane coordinates to prevent the model from learning the features of a non-lane. For each predicted lane, the coordinate is firstly expanded *σ* pixels to avoid the lane falling out of the waiting area, and a region *Φ* whose coordinates are (*x*, *y*, *w* + 2*σ*, *h* + 2*σ*) is obtained. To obtain accurate lane position, it is very important to detect the lane edge in *Φ*. In this study, by using the idea of the adaptive edge detection introduced in [41], the Canny operator is used to detect the lane edge adaptively, and the accurate lane coordinates (*x*’, *y*’, *w*’, *h*’) are obtained automatically. 

First, the lower and upper threshold ($T_{low}$ and $T_{high}$) are set up for the Canny operator to detect the edge points of a lane. Then, the distribution of image grayscale is traversed by the Otsu [42], and an appropriate segmentation threshold is calculated adaptively. The operation is as follows.

The pixels in the image after the non-extremum suppression are divided into three categories (*C*_1_, *C*_2_, and *C*_3_), each of which contains gradient amplitude values defined as $\left\{t_{1},t_{2},\cdots ,t_{k}\right\}$, $\left\{t_{k+1},t_{k+2},\cdots ,t_{m}\right\}$, and $\left\{t_{m+1},t_{m+2},\cdots ,t_{L}\right\}$, where *L* is the number of gradient amplitude. Among these categories, *C*_1_ category denotes a set of the gradient amplitudes of non-edge points in the image, and *C*_2_ and *C*_3_ are the sets of the candidate edge points and the edge points, respectively. The specific steps are as follows.

(1) Calculate the probability distribution of each gradient in the image by:(6)$$p_{i}=\frac{n_{i}}{N},i=0,1,2\cdots ,L,$$
where *N* is the number of pixels in the image, and *n_i_* is the pixel value corresponding to the gradient *t_i_*.

(2) Calculate the gradient amplitude of the whole image by:(7)$$E=\sum_{i=1}^{L}t_{i}p_{i},$$

(3) Calculate the expectations for gradient magnitude $E_{1}\left(k\right)$, $E_{2}\left(k,m\right)$, and $E_{3}\left(m,L\right)$ in *C*_1_, *C*_2_, and *C*_3_ categories, respectively, by:(8)$$\{\begin{array}{l} E_{1}\left(k\right)=\sum_{i=1}^{k}t_{i}p_{i}/\sum_{i=1}^{k}p_{i}\\ E_{2}\left(k,m\right)=\sum_{i=k+1}^{m}t_{i}p_{i}/\sum_{i=k+1}^{m}p_{i}\\ E_{3}\left(m,L\right)=\sum_{i=m+1}^{L}t_{i}p_{i}/\sum_{i=m+1}^{L}p_{i} \end{array},$$

(4) Calculate the variance between classes by:(9)$$\sigma^{2}\left(k,m\right)=\sum_{i=1}^{k}p_{i}{\left[E_{1}\left(k\right)-E\right]}^{2}+\sum_{i=k+1}^{m}p_{i}{\left[E_{2}\left(k,m\right)-E\right]}^{2}+\sum_{i=m+1}^{L}p_{i}{\left[E_{3}\left(m,L\right)-E\right]}^{2},$$
where k∈[1,L] and m∈[k+1,L]. When $\sigma^{2}\left(k,m\right)$ is maximal, the class separability is optimal. By traversing *k* and *m* in their corresponding intervals, when the maximum $\sigma^{2}\left(k,m\right)$ is obtained, then the *k* and *m* at this time are the $T_{low}$ and $T_{high}$, respectively.

The detected lane can be relocated by the adaptive threshold detection algorithm based on the Canny, and accurate position can be obtained. However, due to the low detection accuracy of the model in the first stage, some lane may not be detected. Therefore, if only *T ≤ ξ* is used to judge whether the lane has been detected, the unrecognized lane may be classified as a background, which is extremely disadvantageous to the model training. The *ξ* value of the unrecognized lane is smaller than *T* but still larger than the confidence value of the background. An object with a confidence value of less than *T* may be the lane or other objects. To avoid the interference of an unrecognized lane in model training, the ambiguous object is shielded from the image. First, the bounding boxes with the confidence value in the range (*T*/4, *T*) are expanded to (*x*, *y*, *w* + 2*σ*, *h* + 2*σ*), and then, the pixel value in (*x*, *y*, *w* + 2*σ*, *h* + 2*σ*) is set to 0; thus, a real lane with a threshold less than *T* appears as a black block in the image, which is not very realistic, but in such a way the model training will not be affected because it is classified as the background. In the fine-training process in the second stage, the detection ability of the model is gradually enhanced, and the number of undetected lanes is reduced. By setting the growth factor *ζ* for *T*_0_, the threshold value of the model is raised continuously, and the false detection rate of a lane is reduced. According to the above configurations, the lane features in complex scenarios can be learned adaptively. The model training is given in the following (Algorithm 1). 

**Algorithm 1** The second-stage model fine-training algorithmInitialization:  1. Collect the image sets under various scenarios; 2. First-stage model training *hlane*_1;  3. Set the confidence threshold *T*; 4. Set the threshold growth factor *ζ*;  5. Divide images into multiple sets ***M***, of which each contains the batch size number of images Training process: For batch in ***M***:  for *X_i_* in *batch:*
  1. Obtain the coordinates, width, and height (*x*, *y*, *w*, *h*), and confidence ξ using the model *hlane*_1    if *T* ≤ *ξ_i_*:     (1) Set a new area of the bounding box as (*x*, *y*, *w* + 2*δ*, *h* + 2*δ*)     (2) Determine the lane edge using the adaptive threshold detection algorithm based on the Canny     (3) Binarize the area and determine the connected domain of the lane     (4) Determine a new bounding box (*x*’, *y*’, *w*’, *h*’) of the lane   else if *T*/4 ≤ *ξ_i_* < *T*:     (1) Set a new area of the bounding box as (*x*, *y*, *w* + 2*δ*, *h* + 2*δ*)     (2) Set the pixel value in the new area to 0   end  end  2. Obtain the processed image sets $X=\left\{X_{i}^{\prime },X_{i+1}^{\prime },\ldots ,X_{i+batchsize-1}^{\prime }\right\}$  3. Make *T* = *T* + *ζ*, retrain *hlane*_1end

### 2.4. Lane Fitting

The lane positions obtained by the final lane detection model are independent. Since the goal of this study is to obtain a continuous curve as the model output, the independent blocks are fitted to a line. However, the candidate points that are used to fit the lane are often interference points formed by the off-lane edges. On the other hand, the RANSAC algorithm uses only some of the points to estimate the line parameters, and the existence of the individual outliers has no effect on the final fitting result; thus, the ability of noise suppression is strong. Therefore, the RANSAC algorithm presented in [18] is used in this study. On the basis of the Bezier curve model, the candidate points of the curve are adjusted according to the actual curve to complete the lane fitting.

Considering the complexity of road shapes in real-world road scenarios, the third-order Bezier curve is enough to describe the shape of most lanes such as straight line, parabolic curve or S-shaped curve, etc. Therefore, the third-order Bezier curve is chosen as the model to fit the lane. 

## 3. Experimental and Discussion

The experiment was conducted using the Ubuntu 14.04 operating system, i7-7600K CPU, 16 GB memory, and NVIDIA GTX1080 graphics card. In the existing research on environmental awareness regarding the intelligent vehicles, the effectiveness of the algorithms is generally evaluated on KITTI and Caltech datasets, so these two data sets were also used in our experiment.

This Section consists of the following steps. First, the KITTI and Caltech data sets were used to generate label images. Then, the first-stage YOLO v3 model for lane detection was trained. After the first-stage model is obtained, part of lanes in the image were detected and an adaptive edge detection algorithm was used to relocate the lane, which can be used as label data for the training of the second-stage model. To better illustrate the effectiveness of the proposed algorithm, the current advantage object detection algorithms were introduced for comparison. Furthermore, due to the independent distribution of lane estimated by the YOLO v3 model, the RANSAC was applied to fit the curve of lane.

### 3.1. Model Training

The road images in the raw data of the KITTI traffic dataset amounted to as much as 20.8 GB. This dataset contained the road images in many scenarios, which could provide sufficient training and test data for the needed research. However, almost all lanes in the KITTI dataset were in white color. On the other hand, the Caltech Lanes dataset contained not only white lanes but also yellow ones. In the Caltech Lanes dataset, there were only 1225 images. Since the CNNs usually learn low-level features such as edges and colors in the first few layers, the key features used in object detection and classification can be learned at higher layers. Considering that both KITTI and Caltech datasets denoted the image sets of traffic scenarios, having a certain level of similarity, and the high-level features of the network were also related, it was only necessary to retrain the model trained with the KITTI dataset to detect the yellow lane using the transfer learning. The training process is shown in Figure 3. First, the first-stage KITTI model was trained with the KITTI label images, and then, the KITTI raw images were fed to the model input. After the detection results were optimized according to the training method presented in Section 2, they were used as the label images for the training of the second-stage KITTI model. After that, the Caltech label images were used to train a first-stage Caltech model based on the second-stage KITTI model, and the same method was used to train the second-stage Caltech model with the raw Caltech images. After training, the resulting model could identify both white and yellow lanes. This training process avoids the repetitive training of the primary parameters of the model, making the overall training process simpler.

#### 3.1.1. Evaluation of Label Image Generation Algorithm

The 10.9 GB of (13,855) KITTI’s and all (1225) Caltech raw images are processed according to the method of automatic generation of label images in Section 2.1. The initial values of the parameters for generating the label images were as follows: *l_thresh_min* = 212, *l_thresh_max* = 255, *b_thresh_min* = 135, *b_thresh_max* = 200.

Examples of some label images generated are shown in Figure 4. This method can accurately mark lane location in brightly lit and unobstructed scenarios, as shown in Figure 4aI,aII,bI. However, there are obvious defects in marking the lane only with color information. For example, in Figure 4aIII,bII, they are not marked successfully. Furthermore, the turnoff sign in Figure 4aIV and the cross buck sign and the white vehicle in Figure 4bIII are marked as lane, because their color is similar to the lane. To ensure the accuracy of the training model, it is necessary to remove the poor label images manually. The operation steps are simple. It takes only 1–2 s to eliminate a poor image, while it takes an average of 2 min to generate a label image using the time-slice method. Thus, an enormous amount of time for the training of the deep learning model can be saved. After eliminating the poorer images, 1749 KITTI label images with a size of 1272 × 375 pixels and 193 Caltech label images with a size of 640 × 480 pixels were obtained.

#### 3.1.2. Model Training on KITTI Data Sets

The YOLO v3-based two-stage lane features adaptive learning model proposed in this paper was built with the Keras based on the Tensorflow. The initial values of the training parameters were as follows: maximal number of epochs was set to 40,000 (*max_epoches* = 40,000), learning rate was set to 0.001 (*learning_rate* = 0.001), batch size was set to 128 (*batchsize* = 128), *S* was set to 13 (*S* = 13), and momentum was set to 0.9 (*momentum* = 0.9).

After 33 h of training using the KITTI label images, the first-stage KITTI model was obtained, and the detection results on the images were as shown in Figure 5. The lanes on the road with the bright light and no interference information were detected accurately, such as those in Figure 5a,b. However, as is shown in Figure 5c, only a small number of lanes were detected in the images with overshadowing because the label images used for the model training did not contain any shadow. Similarly, due to the poor generalization ability of the model, some objects similar to the lane at the end of the vehicle and the road markings were also misidentified, as shown in Figure 5IV. Obviously, the first-stage KITTI model obtained had poor detection results in complex scenarios and thus needed further training.

#### 3.1.3. Training of the Second-Stage Model

Although the first-stage KITTI model could not detect all the lanes accurately, it could still detect some of the lanes effectively. By relocating the detected lanes and shielding some possible ones with the method in Section 2.2, the final obtained images can be used as label data for further training. The initial values of the second-stage training parameters were as follows: maximal number of epochs was set to 20,000 (*max_epoches* = 20,000), learning rate was set to 0.0005 (*learning_rate* = 0.0005), batch size was set to 128 (*batchsize* = 128), momentum was set to 0.9 (*momentum* = 0.9), and *σ* was set to 3 (*σ* = 3). The remaining 9.9 GB (12979) unlabeled KITTI images were used to train the second-stage model. To consider the lower error and the missing detection rate, an initial threshold *T*_0_ was set to 0.1 (*T*_0_ = 1). In the second-stage KITTI training, the detection ability of the model was gradually enhanced, and the number of undetected lanes was reduced. Therefore, by setting up the growth factor to 2*e*^−5^ (*ζ* = 2*e*^−5^) for *T*_0_, the detection threshold of the model was constantly raised during the training process.

The training of the second-stage KITTI model for white lane recognition took 113 h. The detection results are shown in Figure 6. The model not only detected the lanes in simple scenarios, but obtained even better results in complex scenarios. In practice, the shadow formed by trees and vehicles on both sides of the road is the most common disturbance in lane detection. To improve the lane detection ability, some scholars have even studied algorithms to eliminate shadow [43,44]. In this study, the second-stage KITTI model did not need to preprocess the shadow, and the detection result was remarkable under various shadow conditions, as shown in Figure 6a,b. Moreover, the model had a strong ability to suppress a strong light under the overexposure conditions, as shown in Figure 6c. Furthermore, the problem of false detection of the vehicle rear, cross buck, and turn off sign was also solved by the second-stage model, as shown in Figure 6d.

Since the images in the KITTI data set are all white lanes, the model trained by the image set can accurately detect white lanes, but not yellow lanes. In the case of yellow lanes, the primary yellow lane detection model was trained using 193 yellow label images from the Caltech dataset based on the existing second-stage KITTI model using the transfer learning. Then, 510 images were randomly selected from the remaining 1032 Caltech images for the training of the second-stage model. Similarly, using the training method given in Section 3, the detection results of the first-stage model were optimized and used as label images so that the second-stage model could be further trained. The maximal number of epochs for the first and second stage were set to 40,000 (*max_epoches* = 40,000) and 80,000 (*max_epoches* = 80,000), respectively. In addition, other parameters for the two stage models were the same as the training parameters on the KITTI dataset.

The detection results of the first- and second-stage Caltech model on the Caltech dataset are shown in Figure 7. As the model was transferred from the KITTI dataset, the ability to detect a white lane was preserved. The first-stage Caltech model, trained with a small number of Caltech label images, can accurately position the white lanes, as shown in Figure 7a. However, for the yellow lanes, since the yellow lane label images in the simple scenario were used in the first-stage Caltech model, the model is unable to detect yellow lanes efficiently. The ability of the second-stage model to recognize a yellow lane was greatly improved. As shown in Figure 7b, the yellow lane could be detected and was no longer affected by the pavement markings.

Although adaptive learning of the lane features can be achieved in various scenarios using this design method, it needs to follow certain steps. During the training of the second-stage model, images were fed to the first-stage model in a sequence from simple to complex in turn, which enabled the model to learn the lane features under various scenarios quickly. However, in the early stage of the training, some lanes might not be detected accurately by the first-stage model if the images of the complex scenarios were used as a direct input, which might easily lead to longer training time. Thus, it was necessary to input the images in order from simple to complex. The scenarios of the images in a folder were almost the same so that we could sort all image folders according to the image complexity before training. This operation was simple and had little impact on the overall training time. Compared with other methods which adopt manual label images training model, the method in this research prevails significantly.

### 3.2. Evaluation of the Effectiveness of Detection Algorithm

To better illustrate the effectiveness of the proposed algorithm, both KITTI road data (http://www.cvlibs.net/datasets/kitti/eval_road.php) and untrained Caltech images were used as the test data. Among them, the KITTI road data contained three types of images, UM (urban marked), UMM (urban multiple marked lanes), and UU (urban unmarked), which were divided into two sets, the test set contains 290 images and the training set contains 289 images. To evaluate the performance of the model more comprehensively, the 579 KITTI images for testing were obtained by combining the above two categories of images. In the Caltech Lanes dataset, 522 images were not used in training. The image annotation tool LabelImg was used to label the two types of images, and the obtained images were used as the ground truth for testing.

In this work, the mean Average Precision (mAP) and detection speed were used as the evaluation criteria. The effectiveness of the proposed algorithm was compared with the current advantage object detection algorithms including the Fast-RCNN [25], Faster-RCNN [26], Sliding window & CNN [23], SSD [28], and Context & RCNN [45]. According to the distribution of lanes in bird-view images, the longitudinal detection density increased. By dividing the images into *S* × 2*S* grids for YOLO v1 [27], YOLO v2 [29], and YOLO v3 [31], the advantages of the structure with *S* × 2*S* was illustrated.

The building of the detection models based on the YOLO v1 and YOLO v2 was the same as that of YOLO v3. They both adopted the Keras framework based on the TensorFlow, and other algorithms that mentioned above used the author’s source code. The training of different lane detection models was completed according to the training procedure using the KITTI and Caltech images presented in Section 3.1. As the final models are all two-stage YOLO, they were abbreviated as T-S YOLO.

Figure 8 showed the result of PR (precision-recall) curve on KITTI and Caltech ground truth. The trend of PR on the two datasets was basically the same. It is evident that the detection algorithm based on the YOLO series had obvious advantages in lane detection, and the model with *S* × 2*S* structure based on the YOLO v3 was the best on the two datasets.

The mAP and detection time of all algorithms are shown in Table 1, where it can be seen that the Sliding window & CNN [20] and Context & RCNN [45] did not resize the image to a certain size, so the larger the size of the image was, the longer the detection time was. The other algorithms resized the images to the same size, so there was a little difference in detection time. 

The Sliding window & CNN [23] used the window sliding method to scan the whole image to obtain lane location. Although it achieved high accuracy on both KITTI and Caltech datasets, the detection took quite a long time. On the other hand, the Fast RCNN [25] obtained the feature map by feeding the whole image to the CNN, which greatly improved the detection speed, but the speed was still slow because of the selective search method used to generate the candidate boxes; the detection time for the images in both KITTI and Caltech datasets was at least 2 s. The Faster RCNN [26] adopted the RPN method to obtain the candidate bounding boxes, both accuracy and detection speed were improved significantly. However, the requirement for real-time detection was still not met, and the accuracy was still not high enough. An important feature, i.e., the context information of the lane, was used to construct a model named Context & RCNN [45] by constructing the network based on the RCNN. The accuracy on the KITTI and Caltech datasets was improved greatly and reached 79.26% and 81.75%, respectively. However, the problem of a large amount of computation still persisted. The object classification and localization by the YOLO [27] were performed within one step, and high-accuracy real-time processing was achieved. The SSD [28] integrated the idea of end-to-end in the YOLO and the mechanism of the anchor box in the Faster RCNN; thus, the accuracy and speed of lane detection were improved significantly. The average image processing time on KITTI and Caltech datasets was 29.3 ms and 25.6 ms, respectively. Furthermore, the YOLO v2 [29] added the anchor boxes on the basis of the YOLO and automatically searched for the best dimension using the K-means clustering, so the detection accuracy was further improved. The YOLO v3 detection model [31] was extended to the Darknet-53 network on the basis of the Darknet-19, so the feature ability extraction was enhanced. Also, by improving the loss function and increasing the number of anchor boxes, the ability of the model to detect lane was further improved.

It can be seen from Figure 8 and Table 1 that the accuracy of lane detection can be significantly improved by dividing the image into *S* × 2*S* grids. Furthermore, this can also illustrate that it is feasible to cut the images according to the distribution characteristics of lanes. However, the time for detection was prolonged due to the growing computing load caused by the increase of detection density. In this study, the lane features in different scenarios were learned by designing an adaptive lane feature learning framework, and the whole process was almost automated. The mAP value of the YOLO v3 on KITTI and Caltech was 88.39% and 89.32%, respectively, and the average processing time of each dataset image was only 25.2 ms and 24.7 ms. This showed that, compared with other training methods that rely on manual labeling of images, the method in this research was simpler to operate, and the results were more reliable.

### 3.3. Lane Line Fitting Results

The lane fitting process is shown in Figure 9. First, the original image (Figure 9a) was fed as an input to the model. After the warp perspective mapping, the bird-view image that filtered out most of the irrelevant information was obtained (Figure 9b). The lane outputs of the YOLO v3 model were independent, but in practice, a continuous line is needed, so the individual detected blocks should be fitted into a curve. To improve the computational efficiency, the irrelevant information outside the lane diagram was first shielded by setting the pixels of non-lane blocks to 0 (Figure 9c), and then, the third-order Bessel curve was employed to fit the lane using the RANSAC method. After obtaining the left and right curves in the bird-view image, the lane was mapped to the original image. Finally, the image with the lane marking was obtained, Figure 9d.

The detection and fitting results of KITTI and Caltech images are shown in Figure 10, where it can be seen that the lanes in the shadow, backlight, and lane wear scenarios were accurately detected using the proposed lane detection method. After fitting the lane blocks using the RANSAC method, the final lane curve was obtained.

## 4. Conclusions

In this research, a two-stage network is designed to learn the lane features in different scenarios automatically, and a model that can detect a lane in complex scenarios is obtained. To train the first-stage model, an automatic method for generating label images using color information was proposed. According to the lane distribution characteristic, the density of the candidate boxes on the *y*-axis is increased to divide the images into *S* × 2*S* grids, making it more suitable for lane detection. Moreover, to simplify the training process, the idea of the adaptive edge detection is employed so that the lane features in complex scenarios can be adaptively and automatically learned. The experimental results show that the final model on the KITTI and Caltech datasets reach a high accuracy. 

However, the spatial relationship of lane is not considered in this study. Therefore, in our future researches, the spatial distribution of lanes will be analyzed to further improve the detection accuracy.

## Figures and Tables

**Figure 1 sensors-18-04308-f001:**
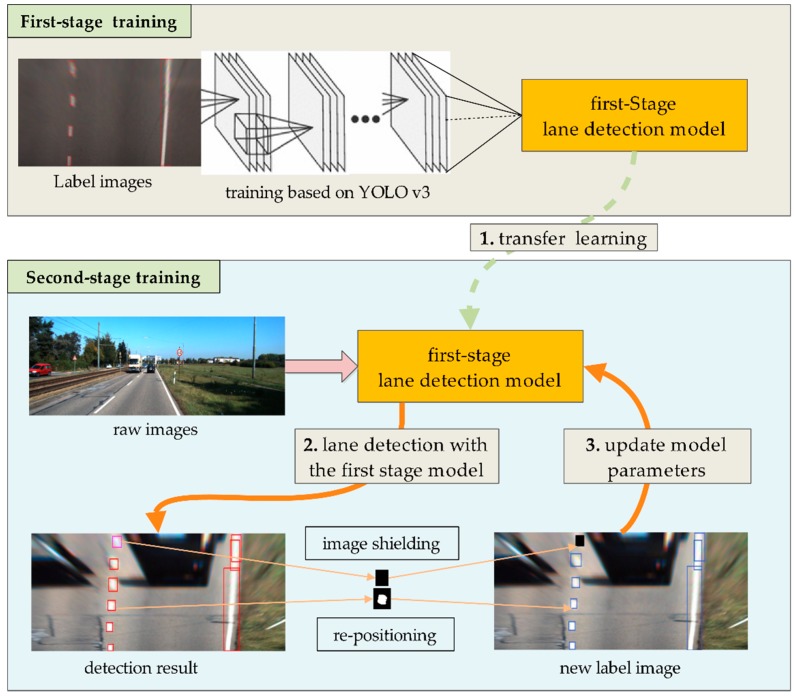
Structure of the two-stage lane detection model based on the YOLO v3.

**Figure 2 sensors-18-04308-f002:**
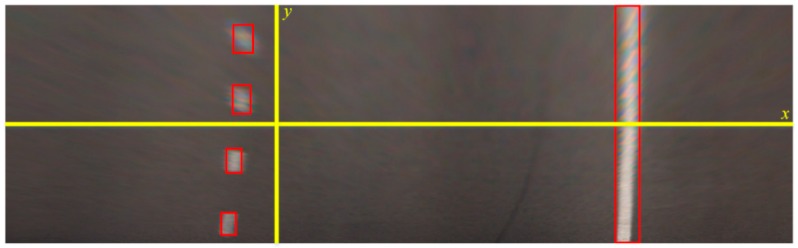
Lane distribution in the bird-view image.

**Figure 3 sensors-18-04308-f003:**
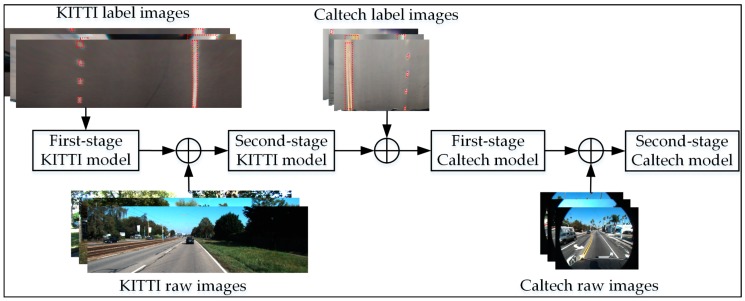
Training flowchart.

**Figure 4 sensors-18-04308-f004:**
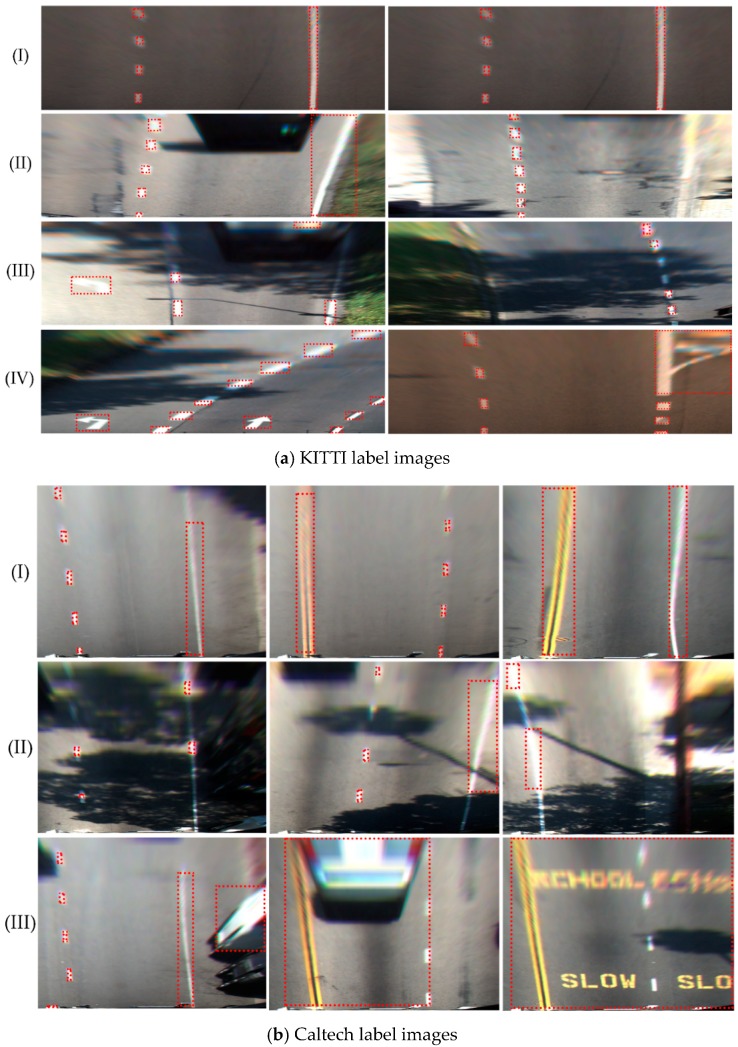
Label images.

**Figure 5 sensors-18-04308-f005:**
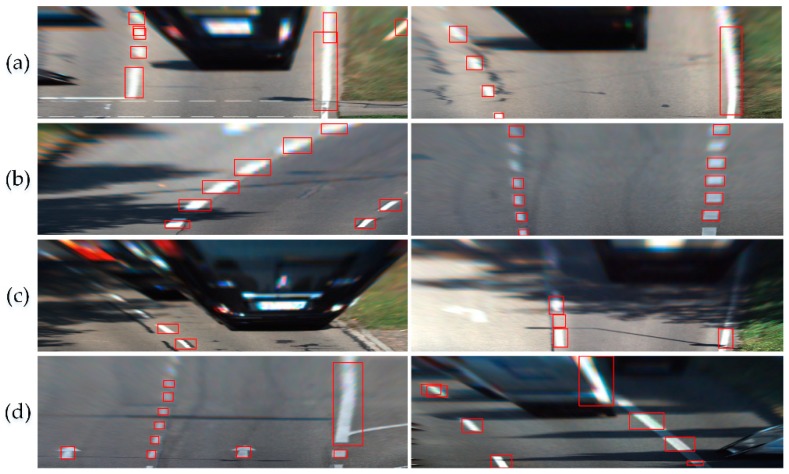
Detection results of the first-stage KITTI model on the KITTI dataset.

**Figure 6 sensors-18-04308-f006:**
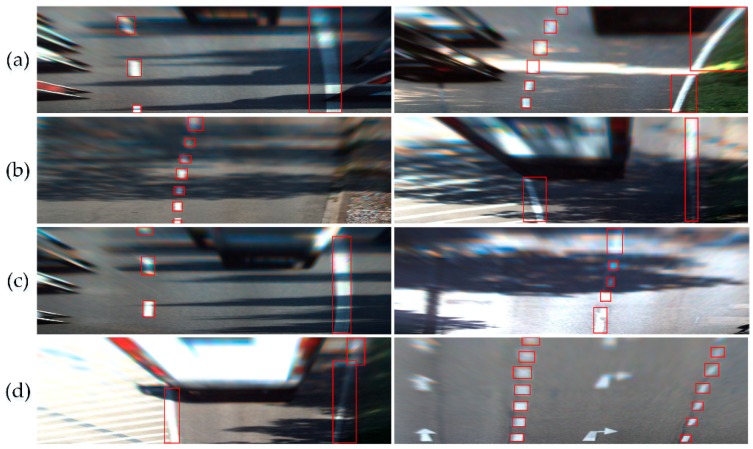
Detection results of the second-stage KITTI model on the KITTI dataset.

**Figure 7 sensors-18-04308-f007:**
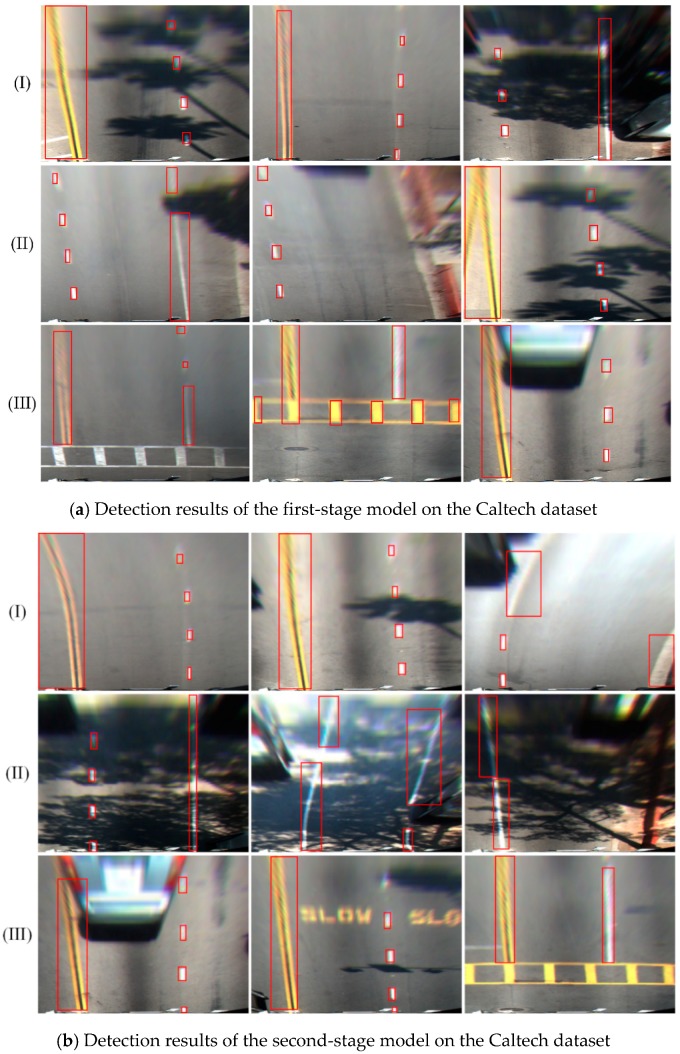
Lane detection results on the Caltech dataset.

**Figure 8 sensors-18-04308-f008:**
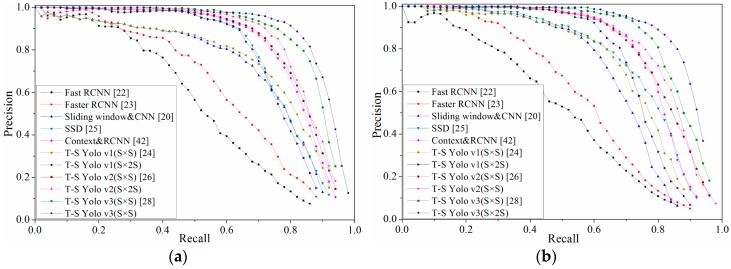
PR curves of all algorithms on the two datasets. (**a**,**b**) are the PR curves on the KITTI and Caltech dataset, respectively.

**Figure 9 sensors-18-04308-f009:**
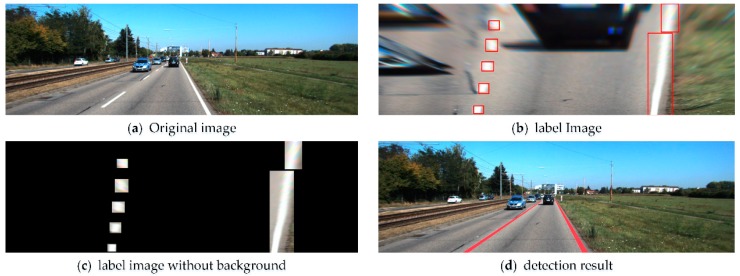
The lane fitting process.

**Figure 10 sensors-18-04308-f010:**
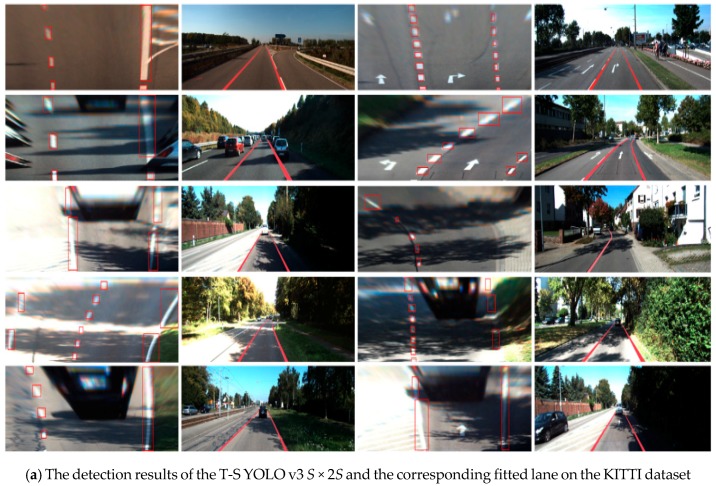
Fitting results after lane detection (the odd columns show the lane detection result under the bird-view perspective, and the even columns show the lane fitting result mapped to the original image).

**Table 1 sensors-18-04308-t001:** Detection accuracy and speed of all lane detection algorithms on KITTI and Caltech datasets.

Algorithm	KITTI		Caltech	
	mAP (%)	Speed (ms)	mAP (%)	Speed (ms)
Fast RCNN [25]	49.87	2271	53.13	2140
Faster RCNN [26]	58.78	122	61.73	149
Sliding window & CNN [23]	68.98	79,000	71.26	42,000
SSD [28]	75.73	29.3	77.39	25.6
Context & RCNN [45]	79.26	197	81.75	136
Yolo v1 (*S* × *S*) [27]	72.21	44.7	73.92	45.2
T-S Yolo v1 (*S* × 2*S*)	74.67	45.1	75.69	45.4
Yolo v2 (*S* × *S*) [29]	81.64	59.1	82.81	58.5
T-S Yolo v2 (*S* × 2*S*)	83.16	59.6	84.07	59.2
Yolo v3 (*S* × *S*) [31]	87.42	24.8	88.44	24.3
T-S Yolo v3 (*S* × 2*S*)	88.39	25.2	89.32	24.7

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
