# Peer review of "A Fast Learning Method for Accurate and Robust Lane Detection Using Two-Stage Feature Extraction with YOLO v3"

_sensors, 2018, doi:10.3390/s18124308_

Round 1
Reviewer 1 Report
Dear Authors:
I regret to inform you that although the paper is well written and is easy to follow. From my point of view the paper does not add a significative content to the state of the art, due to the mayor contribution of the paper is an automatic labeled system based on thresholding the images in order to train an existing deep neural network, that it is not enough to publish the paper in the journal.
Regards
Author Response
Dear reviewer:
We are sincerely grateful for the insightful and constructive comments concerning our manuscript entitled “A fast learning method for accurate and robust lane detection using two-stage feature extraction with YOLO v3”. The comments isvery valuable and helpful for us to revise the manuscript, as well as the important guiding significance to our researches. We have studied the comment and have made correction which we hope meet with approval. In the file of the attachment, we carefully present our response to the comment.

Reviewer 2 Report
The manuscript presents obtain the lane position on a road. A lane size is relatively small compared to the whole image, and the lane spacing becomes shorter after it is converted to the bird-view image. The YOLO v3 has obvious advantages in detecting small target objects regarding the speed and accuracy. Therefore, based on the YOLO v3 framework, a lane detection model is designed, and the lane detection and tracking are completed.
The manuscript should attract an audience in the scientific field of remote sensing applications. The manuscript is written quite good, but the results must be extended. In Introduction, authors explained the proposed used technology, methods and describe in detail their experiments .
However, I have some important remarks to the manuscript:
Introduction is written well, but should be extended to describe the use of YOLO algorithm.
Section 2.1 There is no information about characteristic of the training images example resolution etc.
Line 166: What is the size of the grids ? What is the concrete value of the “S” ?
I have objections to the discussion section. The authors need to re-organize ,the results and discussion therein to better highlight to the reader what was done and what is relevant. The gain of the presented technique for the addressed application should be made more explicit in the form: What do the findings allow what was possible before. Authors should discuss the results and how they can be interpreted in perspective of previous studies and of the working hypotheses.
In general, I think paper requires minor corrections.
The paper also requires minor editorial corrections.
Author Response
Dear reviewer:
We are sincerely grateful for the insightful and constructive comments concerning our manuscript entitled “A fast learning method for accurate and robust lane detection using two-stage feature extraction with YOLO v3”. Those comments are very valuable and helpful for us to revise the manuscript, as well as the important guiding significance to our researches. We have studied comments and have made correction which we hope meet with approval. In the file of the attachment, we carefully present our response to the comments.

Reviewer 3 Report
This paper is well written, and the algorithms it claims are clear. In addition, the experiment and the analysis of the results were also good. However, in order to be accepted, some demerits need to be modified.
If there are several authors of reference, the notation must be revised in its entirety.
for example, Ding [5] ==> Ding et al. [5], El Hajjouji [6] => Hajjouji et al. [6]
2. What is the difference between the RANSAC method used by the author and the reference 'Kim [17]'?
3. 'in the bird-view images, the lane appear in the form of parallel lines, which make~' How does this statement differ from inverse perspective mapping (IPM)? Can you create a parallel image with only a simple affine transform?
4. In Figure 1, please put the number of each algorithm according to the sequence of the proceedings (arrows).
5. 'dividing the image into S×2S grids~' Why is the number of the grid set to sx2s? What if 3 or 4? Also, is not this value dependent on the camera height and affine transform (or IPM)?
6. In Eq. (4), why authors decided lamda coord as 5? Is there any theoretical or experimental basis?
7. At page 5, 'and length and width of the sliding window~'. => width and height of ~?
8. In Eq. (5), there was no explanantion about I^obj_ij and I^nolane_ij
9. In Eq (6), Does L means the gradient amplitude level?
10. In Eq. (9), E3(m) is changable as like E3(L) or E3(m,L)
11. In page 7 algorithm, the first 'if epsilon <=T' and 'else if T/4 < epsilon <=T' is very ambiguous. This part is a duplicate of the condition, is not it wrong?
12.Lines 326 to 339 on page 9 are similar or identical to the above phrases. Delete it.
13.In all sentences, 'Figure.' Modify the notation to the format shown in Figure 1 ~ (remove point).
14. In all experiments, YOLO (SXS) is labeled as YOLO, is your algorithm fit? If so, modify your algorithm, such as two-stage YOLO or T-S YOLO, so that the two-stages are joined together. Table 1 is same.
15. In Table 1, SX2S takes longer in KITTI, but why did Caltech reduce the time in YOLO v3 (Sx2S) by 0.6 more? A detailed explanation is needed.
16. In the conclusion, it is not desirable to number the authors' algorithms. I would like to recommend to modify it to describe the rough introduction, analysis, problems and limitations of your algorithms.
Author Response

(The authors gave the same response as above.)

Round 2
Reviewer 3 Report
All comments are satisfied in the revised manuscript.
Minor check
Kim et al. [19] ==> Kim and Lee [19] (when the number of authors is below 2)
Please check other references.
Author Response
Dear reviewer:
We are very pleased to learn that our manuscript is acceptable for publication in Sensors with minor revision. Our deepest gratitude goes to you for your careful work and thoughtful suggestions that have helped improve this paper substantially. In the following, we carefully present our response to the comment.
Question: All comments are satisfied in the revised manuscript. Minor check. Kim et al. [19] ==> Kim and Lee [19] (when the number of authors is below 2). Please check other references.
Reply : Thank you for pointing this out. After the examination, in addition to reference 19, the same types of problems are references 9, 18 and 33, they all have been corrected in the revised manuscript.